# Orthobiologics Revisited: A Concise Perspective on Regenerative Orthopedics

**DOI:** 10.3390/cimb47040247

**Published:** 2025-04-02

**Authors:** Fábio Ramos Costa, Luyddy Pires, Rubens Andrade Martins, Márcia Santos, Gabriel Silva Santos, João Vitor Lana, Bruno Ramos Costa, Napoliane Santos, Alex Pontes de Macedo, André Kruel, José Fábio Lana

**Affiliations:** 1Department of Orthopedics, FC Sports Traumatology, Salvador 40296-210, Brazil; 2Department of Orthopedics, Brazilian Institute of Regenerative Medicine (BIRM), Indaiatuba 13334-170, Brazil; luyddypires@gmail.com (L.P.); gabriel1_silva@hotmail.com (G.S.S.); dranapolianesantos@gmail.com (N.S.); alex_macedo@icloud.com (A.P.d.M.);; 3Regenerative Medicine, Orthoregen International Course, Indaiatuba 13334-170, Brazil; kruel.andre@gmail.com; 4Medical School, Tiradentes University Center, Maceió 57038-000, Brazil; rubensdeandrade@hotmail.com; 5Nutritional Sciences, Metropolitan Union of Education and Culture, Salvador 42700-000, Brazil; marcinha_mairi@hotmail.com; 6Medical School, Max Planck University Center (UniMAX), Indaiatuba 13343-060, Brazil; jvblana@gmail.com; 7Medical School, Zarns College, Salvador 41720-200, Brazil; fabiocosta7113@gmail.com; 8Clinical Research, Anna Vitória Lana Institute (IAVL), Indaiatuba 13334-170, Brazil; 9Medical School, Jaguariúna University Center (UniFAJ), Jaguariúna 13911-094, Brazil

**Keywords:** orthobiologics, stem cells, growth factors, regenerative medicine, tissue regeneration

## Abstract

At the forefront of regenerative medicine, orthobiologics represent a spectrum of biological substances that offer promising alternatives for tissue repair and regeneration. Traditional surgical treatments often involve significant risks, extended recovery periods, and may not fully restore tissue functionality, creating a strong demand for less invasive options. This paper presents a concise overview of orthobiologics, reexamining their role within the broader landscape of regenerative medicine. Beginning with a brief introduction to orthobiologics, the paper navigates through various types of biological materials and their associated mechanisms of action and clinical applications. By highlighting platelet derivatives, bone marrow-derived products, and processed adipose tissue, among others, it underscores the pivotal role of orthobiologics in prompting biological responses like cellular proliferation, differentiation, and angiogenesis, thereby fostering tissue healing. Furthermore, this paper explores the diverse applications of orthobiologics in orthopedic conditions, outlining their utility in the treatment of bone and soft-tissue injuries. Addressing clinical considerations, it discusses safety profiles, efficacy, patient selection criteria, and emerging challenges. With the limitations of traditional medicine becoming more apparent, orthobiologics offer an innovative and less invasive approach to patient care. Looking forward, this paper approaches future directions in orthobiologics research, emphasizing the need for continued innovation and exploration. Through a concise perspective, this paper aims to provide clinicians, researchers, and stakeholders with a comprehensive understanding of orthobiologics and their evolving role in regenerative medicine.

## 1. Introduction

Orthobiologics are biological materials used to enhance tissue healing and regeneration, particularly in the musculoskeletal system. They are traditionally defined as substances naturally present in the body that accelerate the healing of orthopedic injuries [1]. While many are autologous and biologically derived—such as platelet-rich plasma (PRP), bone marrow aspirate concentrate (BMAC), and adipose-derived products—orthobiologics may also include allogeneic tissues, recombinant growth factors, and synthetic biomaterials engineered to mimic or amplify the body’s natural healing processes [1,2,3,4]. These materials are used extensively in regenerative medicine (Figure 1), where their natural composition allows them to integrate seamlessly into biological systems [1]. Bioproducts such as hyaluronic acid (HA) and autologous solutions like processed peripheral blood, adipose tissue, and bone marrow are among the most widely used biologic materials in the burgeoning field of regenerative medicine [1,5,6,7,8]. As advancements in regenerative medicine continue, the role of orthobiologics has expanded significantly, providing clinicians with new and promising alternatives for managing various musculoskeletal conditions [9]. These bioactive agents have garnered increasing attention for their transformative potential in tissue repair and regeneration, offering potentially less invasive alternative treatments to traditional orthopedic interventions [1,8].

The significance of orthobiologics in tissue regeneration is profound. Unlike conventional therapies that often target symptoms, orthobiologics target the underlying mechanisms of injury, seeking to restore damaged tissues to their pre-injury state [10,11]. By leveraging key molecular components such as growth factors, cytokines, stem cells, and the intrinsic scaffolding properties of some biomaterials, orthobiologics facilitate a multitude of biological processes, including cellular proliferation, differentiation, and tissue remodeling. Collectively, these processes may provide a unique and effective approach in tissue healing [1,6,8]. For instance, growth factors within these biologics stimulate pathways involved in tissue regeneration, while the presence of cytokines helps regulate the inflammatory response, creating an ideal environment for tissue repair [12]. Stem cells, derived from sources like bone marrow and adipose tissue, for example, are particularly crucial, as they can differentiate into various cell types, including osteoblasts, chondrocytes, and fibroblasts, which are essential for regenerating bone, cartilage, and connective tissues [13]. This regenerative capacity of orthobiologics presents a key advantage over traditional therapies, which may only alleviate symptoms without addressing underlying tissue damage [10,14].

Recent research has highlighted the potential of orthobiologics not only in orthopedic settings but also in other areas of medicine, including wound healing, sports injuries, and even plastic surgery [15]. By targeting the body’s natural healing processes, orthobiologics offer a holistic approach to treatment, aligning with the principles of regenerative medicine that emphasize repair rather than replacement [16]. In contrast to pharmacological interventions, which often come with side effects and complications, orthobiologics are derived from the patient’s own biological materials, thus reducing the risk of adverse reactions and enhancing the integration of the treatment into the body [5,17,18].

The increasing demand for minimally invasive and biologically compatible treatments has fueled the growth of orthobiologics within the medical community. The use of orthobiologics is becoming more widespread, with their applications spanning from everyday clinical treatments to advanced surgical procedures [19,20]. Their potential to enhance patient outcomes by accelerating healing, reducing inflammation, and improving tissue regeneration is supported by a growing body of evidence [21]. Over the years, the literature has been able to document evidence of improved recovery times in patients treated with orthobiologics, particularly in cases of tendon injuries, ligament tears, and degenerative joint conditions

This manuscript will deliver a concise overview of orthobiologics, from a regenerative medicine standpoint, highlighting their broad range of applications in orthopedic scenarios. Given the wide array of biologics available, this paper aims to focus on the most commonly utilized ones in clinical practice, namely PRP, HA, BMA/BMAC, and nano-fat, offering insights into their sourcing, processing, and clinical efficacy. Beginning with an examination of the main types of orthobiologics and their sources (Table 1), we delve into some of the mechanisms by which these bioactive agents promote tissue repair and regeneration. Understanding these mechanisms is critical, as they form the foundation of how orthobiologics interact with injured tissues and initiate repair processes at the cellular level. Furthermore, we touch on the therapeutic capability of orthobiologics in orthopedics, providing insights into the conditions treated and procedures performed using these innovative therapies.

By revisiting these concepts, our aim is to accentuate the critical and evolving role of orthobiologics in the continuous pursuit of advancements in regenerative medicine, refreshing readers’ comprehension of orthobiologic potential in tissue healing and regeneration. This review not only consolidates current knowledge but also incorporates recent innovations that have refined orthobiologic applications. These include enhanced bioactive molecule interactions, advancements in delivery technologies that optimize retention and bioavailability, and combinatorial strategies designed to maximize therapeutic efficacy. Furthermore, the increasing influence of regulatory frameworks is shaping their integration into clinical practice, guiding translational efforts toward broader and more standardized applications. As the field continues to evolve, new developments in biologic therapies are expected to enhance the efficacy and scope of orthobiologic applications, paving the way for future innovations in regenerative medicine.

## 2. Methods

Literature was reviewed on electronic databases including PubMed and Google Scholar. The search strategy employed a combination of MeSH terms and free-text keywords to ensure comprehensive retrieval of relevant studies. Keywords included ‘orthobiologics’, ‘regenerative medicine’, ‘platelet derivatives’, ‘bone marrow aspirate’, ‘adipose tissue’, ‘hyaluronic acid’, ‘growth factors’, ‘stem cells’, ‘orthopedics’, and ‘tissue regeneration’. Articles published in English-language journals from 2012 to 2024 were included, prioritizing recent advancements in orthobiologic research. However, older foundational studies were also incorporated when relevant to provide historical context and establish fundamental concepts. Screening involved title and abstract review followed by full-text assessment, with inclusion criteria focusing on studies, reviews, and clinical trials related to orthobiologics and their applications in tissue healing and regeneration. Data extraction synthesized key findings on orthobiologic types, mechanisms of action, and clinical applications, providing a comprehensive overview of the current landscape in orthobiologics research.

## 3. Commonly Used Orthobiologics in Regenerative Medicine

Many orthobiologic solutions are employed in routine regenerative medicine procedures. Popular examples include the following: platelet derivatives obtained from peripheral blood, such as platelet-rich plasma (PRP) and platelet-rich fibrin (PRF) [6,15]; hyaluronic acid [5,22]; bone marrow-derived products such as bone marrow aspirate concentrate (BMAC), bone marrow aspirate (BMA), and “hybrid” (combination with other orthobiologics) [1,23,24]; and adipose tissue-derived materials, including macro-fat, micro-fat (MFAT), nano-fat, and stromal vascular fraction (SVF) [7,21].

Platelet Derivatives from Peripheral Blood: PRP and PRF are derived from the patient’s own blood through a process of centrifugation. PRP is rich in growth factors that promote healing and tissue regeneration, while PRF includes fibrin to provide a scaffold for cell migration and growth [6,15]. Numerous systematic reviews and meta-analyses have validated the clinical efficacy of PRP and PRF, particularly in the management of osteoarthritis, tendinopathies, and soft tissue repair, confirming their value as evidence-based orthobiologics [25,26,27,28,29].Hyaluronic Acid: HA is a naturally occurring substance found in connective tissues and joint fluid. It is commonly used as a viscosupplement to lubricate joints, particularly in the treatment of osteoarthritis, improving joint function and reducing pain [5,22]. Meta-analytical evidence supports the effectiveness of HA in reducing pain and improving function in osteoarthritis, further endorsing its widespread clinical adoption [30,31].Bone Marrow-Derived Products: both BMA and BMAC are obtained from the patient’s bone marrow, typically from the posterior superior iliac crest. BMAC is a concentrated form rich in mesenchymal and hematopoietic stem cells (MSCs and HSCs), and growth factors that support tissue repair and regeneration. Despite limitations regarding stem cell count and differentiation capability, bone BMAC still remains a rich source of various regenerative components [32]. In addition to MSCs and HSCs, it also carries megakaryocytes, platelets, growth factors, and cytokines [1,32]. These elements collectively contribute to substantial paracrine effects, which enhance tissue repair and regeneration [1,32]. The presence of these bioactive molecules not only supports cellular proliferation and differentiation, but also modulates the local inflammatory response and promotes angiogenesis, making BMAC a valuable tool in regenerative medicine and orthopedics [32]. Hybrid BMAC may combine BMAC with other orthobiologics, such as PRP, to enhance regenerative potential via synergism [1,23,24]. Systematic reviews have highlighted the therapeutic potential of BMAC in orthopedic applications, including cartilage repair, osteochondral defects, and non-union fractures, with positive clinical outcomes and safety profiles [33,34,35,36].Adipose Tissue-Derived Materials: these biologic materials are all obtained through liposuction and processing techniques. Macro-fat and MFAT provide structural support and cushioning, while nano-fat and SVF are rich in various cells, such as adipose-derived stem cells (ADSCs), mesenchymal and endothelial progenitor cells, lymphatic cells, pericytes, leukocyte subtypes, and vascular smooth muscle cells that promote tissue regeneration and healing [7,21,37]. Recent meta-analyses and systematic reviews have demonstrated the clinical utility of SVF and related adipose-derived products in regenerative medicine, particularly in joint preservation and soft tissue reconstruction [38,39,40].

## 4. Summary of Mechanisms of Action

Orthobiologics are instrumental in promoting tissue repair and regeneration by leveraging the body’s innate healing mechanisms (Figure 2). Stem cells derived from sources like bone marrow and processed adipose tissue are central to this process, as they possess the capability to differentiate into various cell types crucial for tissue regeneration, such as osteoblasts, chondrocytes, and fibroblasts [41]. Studies have demonstrated that MSCs within orthobiologic preparations retain their viability and differentiation potential post-processing [42,43]. This is confirmed by flow cytometry analysis, colony-forming unit assays, and multilineage differentiation tests, including adipogenic, osteogenic, and chondrogenic assays [42,43]. Additionally, ex vivo expansion studies indicate that these cells can remain metabolically active for extended periods, with cytokine signaling playing a role in maintaining their regenerative properties [44]. These findings reinforce the ability of MSCs to migrate to injury sites, proliferate, and contribute to the formation of new, healthy tissue, supporting their therapeutic utility in regenerative applications [45]. Another key mechanism facilitated by some orthobiologic products is the release of a myriad of growth factors and bioactive molecules. Products like PRP and PRF are rich in a variety of growth factors like platelet-derived growth factor (PDGF), transforming growth factor-beta (TGF-β), and vascular endothelial growth factor (VEGF), to name a few. These bioactive molecules (Table 2) stimulate essential cellular activities such as proliferation, angiogenesis, and extracellular matrix synthesis, crucial for tissue repair and regeneration [8,46].

However, their effects are highly context-dependent, regulated by biochemical and biomechanical signals within the local microenvironment. Biochemical, mechanical, and inflammatory factors play essential roles in guiding the differentiation of MSCs within orthobiologic formulations [47]. Biochemical regulation is driven by the interaction of growth factors with key signaling pathways such as TGF-β, bone morphogenetic proteins (BMPs), and Wnt proteins, which collectively direct lineage commitment [48]. For instance, BMP-2 is known to drive osteogenesis [49], whereas TGF-β enhances chondrogenesis by promoting SOX9 expression, a key transcription factor involved in cartilage formation [50].

Beyond biochemical signals, mechanotransduction plays a fundamental role in MSC differentiation by responding to the physical properties of the extracellular matrix and the mechanical forces exerted within the tissue microenvironment [51]. MSCs that are exposed to stiff, high-tension environments, such as those found in bone matrices, tend to favor osteogenic differentiation, while softer, low-tension environments encourage chondrogenic or adipogenic differentiation pathways [52,53,54].

In addition to biochemical and mechanical influences, the inflammatory milieu significantly impacts tissue-specific differentiation [55]. Anti-inflammatory cytokines such as IL-10 and TGF-β promote chondrocyte differentiation, ensuring proper cartilage regeneration [56,57]. Conversely, pro-inflammatory cytokines such as TNF-α and IL-6 can disrupt this balance, potentially driving fibrotic responses and impairing regenerative outcomes if not properly regulated [58,59]. Together, these factors establish a highly dynamic regulatory system that ensures MSCs within orthobiologics differentiate in a manner that aligns with the specific needs of the target tissue.

When it comes to pain relief, researchers have also focused on interleukin-1 receptor antagonist (IL-1Ra), a significant cytokine present in bone marrow-derived products. IL-1Ra acts as a competitive antagonist, binding to the IL-1B and IL-1a isoforms of cell surface receptors, thereby inhibiting IL-1-induced catabolic reactions and inflammatory effects [60]. This also means that IL-1Ra holds significant potential in mitigating matrix degradation. IL-1B is known to stimulate the expression of MMP-3 and TNF-α, promote the secretion of prostaglandin E2, induce chondrocyte apoptosis, and inhibit collagen deposition. By acting as a competitive antagonist to IL-1B, IL-1Ra can effectively counter these catabolic and inflammatory processes, thereby preserving the integrity of the extracellular matrix which is also essential for tissue health [61].

Orthobiologics also often incorporate scaffolds to provide structural support for new tissue growth. These biological scaffolds, whether natural or synthetic, offer a framework for cells to attach, migrate, and proliferate, facilitating organized tissue formation, particularly in complex tissues like cartilage and bone [1,6,62].

Furthermore, these materials regulate inflammation, creating an optimal healing environment by reducing excessive inflammation and promoting a balanced immune response [8]. Enhanced blood supply is also facilitated by these products through growth factor-induced angiogenesis, ensuring improved oxygen, nutrient, and immune cell delivery to the injury site [63].

Cell signaling is very potent in these materials, especially via secretion of cytokines and other bioactive molecules to coordinate the activities of different cell types involved in the healing process [1,64]. Lastly, fibrinolytic activity and tissue remodeling, which involves breaking down damaged matrix components and replacing them with new ones, is supported by enzymes and molecules released by orthobiologics, ultimately restoring normal tissue structure and function [8].

In essence, orthobiologics offer a multifaceted approach to tissue repair and regeneration, encompassing cellular regeneration, growth factor release, scaffold support, inflammation modulation, enhanced blood supply, cell signaling, and tissue remodeling. By harnessing these natural processes, orthobiologics provide an effective and holistic method for healing and regenerating damaged tissues.

## 5. Applications

Orthobiologics have found extensive applications in the field of orthopedics (Table 3 due to their ability to enhance the body’s natural healing processes. These biological substances are utilized to treat a variety of musculoskeletal conditions, offering a promising alternative to operative interventions and conventional management strategies that are mainly focused on the repair of bone fractures, cartilage loss, tendon and ligament injuries, and soft tissue regeneration.

### 5.1. Repair of Bone Fractures

Bone healing is a complex, highly regulated sequence of events that restores injured bone to its pre-fracture condition. The healing process can be categorized into three overlapping phases: inflammation, repair, and remodeling [65,66]. Platelet derivatives are rich in a plethora of growth factors (Table 2) which are essential for the initial inflammatory response and subsequent healing phases. Injecting platelet concentrates like PRP into the fracture site enhances the recruitment and proliferation of cells necessary for bone repair, thereby accelerating the healing process [67]. Bone marrow-derived products like BMA/BMAC contain a mixture of stem and progenitor cells (MSCs and HSCs) which, along with their associated rich secretome and stabilized cellular activity, further support bone regeneration. These components enhance the repair of bone fractures by promoting osteogenesis and angiogenesis, leading to faster and more robust bone healing [68,69].

### 5.2. Cartilage Repair and Regeneration

HA is commonly used to treat chronic and degenerative joint conditions such as osteoarthritis. It improves joint lubrication, reduces friction, and absorbs shocks, thereby alleviating pain and improving mobility [5,22]. HA also supports chondrocytes by creating a conducive environment for their survival and function, facilitating the synthesis of new extracellular matrix components essential for cartilage repair [70]. Additionally, HA’s anti-inflammatory properties reduce local inflammation, protecting cartilage from further degradation [5,22]. MSCs from sources like BMA/BMAC or SVF can differentiate into cells from chondrogenic lineage and produce extracellular matrix components, which are crucial for cartilage repair and regeneration [1,7,37,71]. These cells can therefore be injected into damaged cartilage areas in cases of patellar chondropathy, for example, where they may orchestrate the regenerative cascade and modulate inflammation to enhance repair and improve joint function [7].

### 5.3. Tendon and Ligament Injuries

PRP is extensively used to treat tendon and ligament injuries due to its high concentration of growth factors that stimulate the healing of these tissues [72,73]. PRP injections promote collagen synthesis and remodeling, which are essential for the recovery of ligaments and the biomechanical function of tendons [74]. PRF, in turn, not only displays similar effects but also possesses a robust, porous fibrin matrix that acts as a natural scaffold, facilitating the attachment and growth of cells [6]. This fibrin matrix provides a three-dimensional structure that supports the sustained release of growth factors, which are critical for tissue regeneration [6]. The unique properties of PRF make it particularly effective in the treatment of tendon and ligament injuries. It promotes healing by creating a favorable microenvironment for cell proliferation and differentiation, enhancing the regenerative process [6]. Additionally, the structural support provided by the fibrin matrix helps in stabilizing the injury site, reducing inflammation, and accelerating the recovery process [8]. PRF’s ability to gradually release growth factors over time ensures prolonged stimulation of the healing tissues, making it a valuable tool in regenerative medicine and orthopedics [6].

### 5.4. Soft Tissue Regeneration

Orthobiologics play a significant role in the regeneration of soft tissues, leveraging biological materials that support and enhance the body’s natural healing processes. Key orthobiologics used in soft tissue regeneration (especially in the field of aesthetic medicine) include but are not limited to ADSCs and HA [75,76,77].

SVF-derived ADSCs, like BMSCs, exhibit trilineage differentiation potential, meaning they can differentiate into adipocytes, chondrocytes, and osteoblasts, among other cell types [77]. While their adipogenic differentiation capacity is well documented due to the presence of pro-adipogenic factors in adipose tissue, ADSCs can also contribute to cartilage and bone repair when exposed to appropriate biochemical and mechanical cues [78]. This versatility makes them a valuable tool not only for soft tissue regeneration but also for broader applications in regenerative medicine [7,77]. They secrete a range of bioactive molecules, including growth factors, cytokines, and extracellular vesicles, which modulate the local environment by promoting angiogenesis, reducing inflammation, and stimulating the proliferation and migration of resident cells to the injury site [79]. Clinically, ADSCs can be used to treat muscle injuries, skin wounds, and other soft tissue defects, making them valuable in reconstructive and aesthetic medicine [21,77,80].

HA retains water to maintain tissue hydration, essential for cellular activities and overall tissue health, and provides structural support to the extracellular matrix, creating an optimal environment for cell proliferation and migration [81]. Its significant anti-inflammatory effects help reduce swelling and pain in injured tissues, modulating the inflammatory response to prevent excessive scar formation and promote more organized tissue repair [82]. This is vital for proper epithelial tissue integrity, for example [83,84]. HA is used in various medical treatments, including wound care for skin ulcers, surgical wounds, and burns, enhancing tissue hydration and providing a supportive matrix, making it a key component in soft tissue regenerative therapies [31,85,86]. Synergistically, ADSCs and HA may significantly improve soft tissue regeneration by leveraging their biological properties to accelerate healing and enhance tissue quality and functionality.

## 6. Delivery Systems

A critical aspect of orthobiologic therapies is the delivery of these biological substances to the injured tissues. The effectiveness of orthobiologics largely depends on the precision and efficiency of their delivery methods. Current clinical practices utilize several approaches, ranging from direct injections to more advanced systems like scaffolds and hydrogels [87]. For example, PRP and stem cells are often delivered via local injection directly into the injury site, ensuring a concentrated application [88]. In contrast, emerging technologies such as nanoparticle-based systems and biocompatible scaffolds are being explored for their ability to provide sustained release of bioactive molecules over time, enhancing the regenerative process [89,90]. Scaffold-based delivery systems, particularly in combination with orthobiologics, provide not only structural support but also create an optimal environment for tissue regeneration [6,91]. These systems are tailored to promote prolonged activity of the biologic agents, reduce inflammation, and improve the overall success of the treatment [6,91]. Hydrogels, which act as delivery carriers, are also being studied for their ability to control the release of growth factors and cells, ensuring that the therapeutic agents remain active for a longer duration at the injury site, further optimizing tissue repair [92,93].

## 7. Current Limitations of Orthobiologics

Despite the growing interest and application of orthobiologics, there are limitations that still need to be addressed for wider clinical use. One of the key challenges is the inherent variability in product composition. Since orthobiologics are often derived from autologous sources such as the patient’s blood or tissue, factors like age, overall health, and genetic differences can logically influence the concentration of growth factors, cytokines, and cells [94]. This variability makes it difficult to achieve consistent therapeutic outcomes across different patient populations [95]. In addition, the protocols for preparing and administering these products can vary between practitioners, further complicating efforts to standardize treatments and ensure reproducible results [95].

Another significant limitation is the current lack of long-term clinical studies that comprehensively evaluate the efficacy and safety of orthobiologic therapies [96]. While early results from small-scale trials are encouraging, large-scale, long-term studies are still needed to validate their effectiveness across a variety of conditions and demographics [96]. This gap in evidence may hinder broader acceptance of orthobiologics, particularly in regulatory environments that demand rigorous clinical data. Additionally, the high cost of these treatments can pose financial barriers for patients, limiting access to this innovative care, especially in regions where healthcare systems may not cover these advanced therapies [96].

Furthermore, while the field of orthobiologics is advancing rapidly, there are still unresolved questions regarding optimal dosages, delivery methods, and treatment combinations. Identifying the right biologic agent for the right condition, as well as determining the ideal timing and frequency of treatment, remains an area of active investigation. Addressing these limitations through ongoing research and collaboration between clinicians, researchers, and regulatory bodies will be essential for the continued evolution of orthobiologic therapies.

In addition to the limited availability of large-scale randomized controlled trials, the clinical use of orthobiologics is not without risks. While regenerative outcomes are generally favorable, certain biological actions can have unintended consequences. For example, uncontrolled extracellular matrix remodeling may result in excessive fibrosis, potentially leading to impaired tissue elasticity, restricted joint mobility, or altered biomechanical properties [97]. Moreover, variability in biologic content, including the concentration of cytokines, growth factors, and stem cells, may yield inconsistent therapeutic responses or provoke unpredictable inflammatory reactions [98]. In stem-cell-based applications, concerns have also been raised regarding the potential for undesired differentiation pathways if local signaling cues are dysregulated, which could lead to tumor formation or other adverse effects [99]. These limitations underscore the importance of refining standardized protocols for processing, dosing, and delivery to ensure safety and efficacy. Ongoing efforts to develop evidence-based guidelines and tighter regulatory frameworks will be critical to mitigate risks and optimize the therapeutic profile of orthobiologic interventions.

Nevertheless, orthobiologics play a crucial role in enhancing the repair and regeneration of various musculoskeletal tissues. By leveraging their biological properties, orthobiologics not only mimic the natural healing processes but also improve the efficiency and quality of tissue repair, offering significant benefits in the treatment of bone fractures, cartilage loss, tendon and ligament injuries, and soft tissue regeneration. These advancements highlight the potential of orthobiologics to revolutionize orthopedic and regenerative medicine.

## 8. Future Directions

The field of regenerative medicine and orthobiologic therapy is continuously evolving, driven by ongoing research and technological advancements. As the understanding of cellular and molecular biology deepens, the potential for orthobiologics to transform medical treatments continues to grow. Researchers are exploring novel approaches to enhance the efficacy of these therapies, leveraging cutting-edge technologies and scientific discoveries to push the boundaries of what is possible in tissue repair and regeneration.

### 8.1. Emerging Trends and Developments in Orthobiologics Research

Orthobiologics research is increasingly focusing on optimizing the efficacy and application of these biological substances. One of the emerging trends is the refinement of combinatorial therapies, where multiple orthobiologic agents are used together to synergistically enhance tissue repair and regeneration [22,23,100,101,102,103]. For example, combining MSC-derived stem cells with a growth factor mixture and a stable PRF or HA hydrogel matrix may amplify the regenerative effects compared to using each agent alone [100]. These combination therapies are designed to harness the unique strengths of each biologic, creating a more potent and comprehensive approach to tissue healing.

Additionally, the use of gene-editing technologies, such as CRISPR/Cas9, is gaining traction. CRISPR offers precise control over genetic modifications, allowing scientists to enhance the regenerative potential of stem cells by editing specific genes involved in tissue repair [104]. These technologies allow for the genetic modification of cells to enhance their regenerative capacities, making them more effective in repairing damaged tissues [104]. Researchers are exploring ways to engineer stem cells and other regenerative cells to express higher levels of therapeutic proteins or to resist inflammatory signals, thereby improving their functionality and longevity in clinical applications [105,106]. The continued expansion of gene therapy into the field of orthobiologics holds promise for creating customizable, patient-specific therapies that deliver superior outcomes.

### 8.2. Advancements in Technology and Treatment Strategies

The development of 3D bioprinting represents a significant technological advancement in orthobiologics. This technology allows for the creation of complex, patient-specific scaffolds that can be infused with orthobiologic agents [107,108]. These scaffolds can mimic the natural extracellular matrix, providing the necessary structural support and enhancing the integration and function of transplanted cells [109,110]. Three-dimensional bioprinting has the potential to revolutionize the field by enabling the fabrication of highly customized tissue constructs tailored to individual patient needs [111]. Another promising area is the improvement of delivery systems for orthobiologics. Innovations such as nanoparticle-based delivery systems and hydrogels can provide controlled, sustained release of orthobiologic agents at the injury site [112,113,114]. These technologies ensure that therapeutic agents are delivered precisely where needed, reducing systemic exposure and enhancing the local healing response. Therefore, this alternative approach may extend therapeutic effects and reduce the need for multiple interventions, improving patient comfort and clinical outcomes.

### 8.3. Personalized Medicine and Predictive Analytics

The integration of personalized medicine approaches is set to revolutionize the application of orthobiologics. By utilizing genetic, proteomic, and metabolomic data from individual patients, treatments can be tailored to their specific biological characteristics [115]. This personalized approach can enhance the efficacy of orthobiologic therapies and minimize adverse effects, leading to better patient outcomes [115]. Moreover, the use of predictive analytics and artificial intelligence (AI) in treatment planning is becoming increasingly prevalent. AI algorithms can analyze vast amounts of clinical data to predict how patients will respond to specific orthobiologic treatments, allowing clinicians to make more informed decisions and optimize treatment strategies [116,117]. As AI and predictive analytics continue to evolve, the ability to fine-tune orthobiologic therapies for individual patients will likely become a cornerstone of regenerative medicine.

### 8.4. Regulatory and Ethical Considerations

As orthobiologics continue to advance, it is crucial to address new regulatory and ethical considerations that arise. Establishing robust regulatory frameworks is essential to ensure the safety, consistency, and quality of orthobiologic products [9,118]. Given the complex nature of biologically derived materials, regulatory agencies must establish clear guidelines that govern the production, handling, and use of orthobiologics to safeguard patient health [9]. For instance, in the United States, the Food and Drug Administration (FDA) classifies orthobiologics under a variety of categories, including human cells, tissues, and cellular and tissue-based products (HCT/Ps) [9]. However, the regulation of orthobiologics is not always straightforward, as many treatments involve the manipulation of human tissue, which can introduce additional regulatory hurdles. The FDA’s role is to ensure that these products meet stringent safety and efficacy standards before being approved for clinical use [119].

In contrast, the European Medicines Agency (EMA) takes a slightly different approach, focusing on the development of advanced therapy medicinal products (ATMPs), which include gene therapy, somatic cell therapy, and tissue-engineered products [120]. This classification system allows for a more specific regulatory framework for orthobiologics, but can also lead to delays in approval as new therapies are assessed under strict guidelines. The differences between the FDA and EMA highlight the complexities of global regulation, where different regions may have varying timelines, requirements, and approval processes for biologic products.

Ethical guidelines must also be developed to address issues related to the sourcing of biological materials, patient consent, and the equitable distribution of these advanced therapies [9,121]. Such guidelines are critical in ensuring that orthobiologics are developed and applied in a manner that ascertains patient rights, supports equitable access, and minimizes the risk of exploitation or misuse.

Moreover, as stem cells and other regenerative agents are increasingly utilized, ethical concerns surrounding their manipulation and application in clinical settings will need to be addressed. In addition to establishing ethical standards for cell sourcing and treatment application, further debates will be necessary to navigate the evolving ethical landscape in regenerative medicine.

## 9. Conclusions

In summary, this paper offers a comprehensive overview of the role and applications of orthobiologics in regenerative medicine. By leveraging biological substances such as PRP, PRF, HA, BMA/BMAC, and SVF, orthobiologics enhance the body’s natural healing processes. These biomaterials have demonstrated significant potential in treating musculoskeletal conditions, including bone fractures, cartilage loss, tendon and ligament injuries, and soft tissue regeneration. Their use has become increasingly prevalent in clinical settings, where orthobiologics are recognized not only for their ability to accelerate recovery but also for their long-term benefits in tissue restoration and functional improvement. In contrast, traditional surgical and pharmacological treatments often involve prolonged recovery times, risks of complications, and may fail to fully restore tissue functionality, underscoring the need for more effective alternatives like orthobiologics.

Orthobiologics are pivotal in advancing regenerative medicine, offering promising alternatives to conventional surgical and pharmacological interventions. Their ability to improve tissue repair and regeneration through various mechanisms, such as cell proliferation, differentiation, angiogenesis, and ‘inflammomodulation’ (inflammatory modulation), underscores their transformative potential in clinical and research settings. The integration of orthobiologics into treatment protocols not only accelerates healing but also enhances the quality and functionality of the repaired tissues, ultimately improving patient outcomes. This benefit is particularly valuable in time-sensitive fields like sports medicine, where injured elite athletes require rapid recovery to return to competition as soon as possible. Traditional treatments may not offer the same rapid recovery benefits, making orthobiologics an essential innovation for improving the speed and quality of healing.

Moreover, the ability of orthobiologics to reduce post-surgical complications and minimize the need for long-term medications makes them an attractive option for patients seeking less invasive treatments.

A key contribution of this manuscript is the exploration of how specific orthobiologics—PRP, PRF, HA, BMAC, and SVF—each play unique roles in the repair and regeneration of different types of tissues. By summarizing current applications and mechanisms of action, this paper provides a concise update for understanding the practical uses of these biological substances in orthopedics and beyond. Additionally, this paper highlights the importance of emerging research and technological advancements in orthobiologics, offering insights into the future potential of these therapies. These advancements, including the refinement of combinatorial therapies and gene editing techniques, are poised to expand the therapeutic scope of orthobiologics, enhancing their effectiveness and applicability across a wider range of medical fields. By presenting the latest developments and identifying key areas for future research, this manuscript emphasizes the continuous evolution of orthobiologics and their ever expanding role in regenerative medicine.

Overall, this manuscript underscores the importance of continued research and innovation in orthobiologics, paving the way for enhanced therapeutic strategies that can significantly impact the future of regenerative orthopedics and clinical care. As conventional treatments become more limited in their effectiveness, orthobiologics represent a critical advancement in achieving more reliable, efficient, and patient-friendly outcomes. With ongoing advancements in biomaterials, personalized medicine and delivery technologies, the future of orthobiologics looks promising, with the potential to transform how injuries and degenerative conditions are treated across multiple medical disciplines.

## Figures and Tables

**Figure 1 cimb-47-00247-f001:**
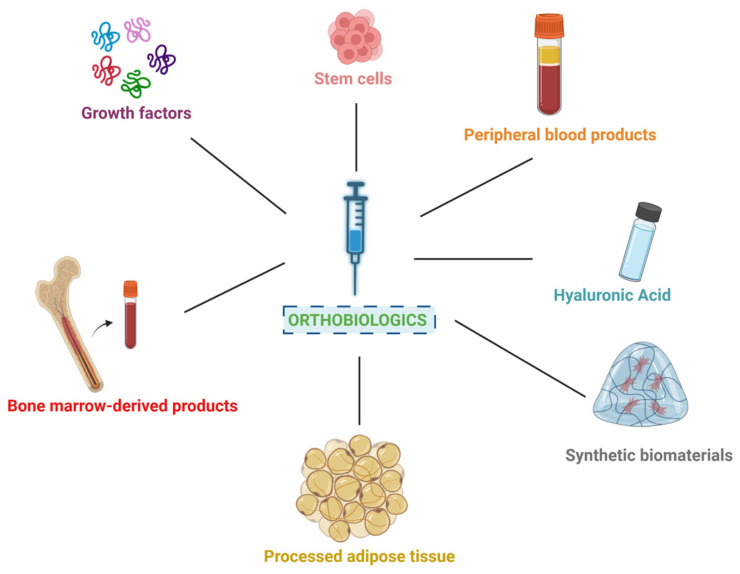
Overview of orthobiologic sources.

**Figure 2 cimb-47-00247-f002:**
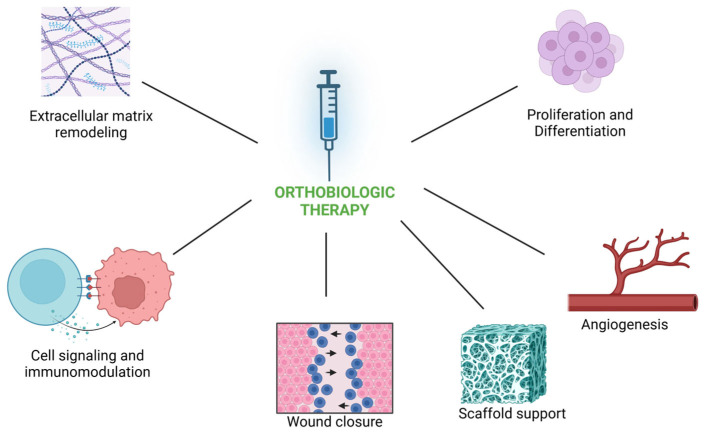
Mechanisms of action of orthobiologics.

**Table 1 cimb-47-00247-t001:** Types of orthobiologics and their characteristics.

Orthobiologic Source	Main Components	Characteristics
Peripheral Blood	Growth Factors (GFs)	PRP, PPP, PRF; Prepared with or without anticoagulant (ACD)
Bone Marrow	Stem Cells (SCs), Growth Factors (GFs)	BMAC, BMA, Hybrid BMAC; May or may not be centrifuged
Adipose Tissue	Stem Cells (SCs)	Macro-FAT, MFAT, Nano-FAT, SVF; Derived from fat tissue, varying in cluster size

Abbreviations: PRP: Platelet-Rich Plasma; PPP: Platelet-Poor Plasma; PRF: Platelet-Rich Fibrin; ACD: Anticoagulant Citrate Dextrose; BMAC: Bone Marrow Aspirate Concentrate; BMA: Bone Marrow Aspirate; MFAT: Microfragmented Adipose Tissue; Macro-FAT: Macroscopic Fat Tissue; Nano-FAT: Nano-fat Tissue; SVF: Stromal Vascular Fraction.

**Table 2 cimb-47-00247-t002:** Key bioactive molecules in platelet-rich plasma (PRP) and platelet-rich fibrin (PRF).

Bioactive Molecule	Biological Role
Platelet-Derived Growth Factor (PDGF)	Promotes cell proliferation and angiogenesis
Transforming Growth Factor-Beta (TGF-β)	Regulates cell growth, proliferation, differentiation, and apoptosis
Vascular Endothelial Growth Factor (VEGF)	Stimulates angiogenesis and increases vascular permeability
Epidermal Growth Factor (EGF)	Promotes cell growth, proliferation, and differentiation
Insulin-Like Growth Factor (IGF)	Stimulates growth and development of cells
Fibroblast Growth Factor (FGF)	Promotes cell growth, proliferation, and differentiation
Hepatocyte Growth Factor (HGF)	Stimulates cell growth, motility, and angiogenesis
Connective Tissue Growth Factor (CTGF)	Promotes the proliferation and differentiation of fibroblasts
Keratinocyte Growth Factor (KGF)	Stimulates epithelial cell growth and differentiation
Platelet Factor 4 (PF4)	Modulates inflammation and wound healing
Interleukin-1 (IL-1)	Plays a role in inflammation and immune responses
Interleukin-8 (IL-8)	Promotes chemotaxis and angiogenesis

**Table 3 cimb-47-00247-t003:** Applications of orthobiologics in clinical contexts.

Orthobiologic	Primary Components	Key Clinical Applications
Platelet-Rich Plasma (PRP)	Growth factors (PDGF, TGF-β, VEGF, EGF)	Tendinopathies, osteoarthritis, muscle injuries, post-surgical healing
Platelet-Rich Fibrin (PRF)	Platelets, fibrin matrix, leukocytes	Wound healing, periodontal regeneration, soft tissue repair
Hyaluronic Acid (HA)	Viscous glycosaminoglycan	Osteoarthritis (intra-articular injection), joint lubrication
Bone Marrow Aspirate (BMA)/Bone Marrow Aspirate Concentrate (BMAC)	MSCs, hematopoietic cells, growth factors	Cartilage repair, bone defects, non-union fractures
Stromal Vascular Fraction (SVF) from Adipose Tissue	MSCs, pericytes, extracellular matrix components	Soft tissue regeneration, wound healing, degenerative joint disease

## Data Availability

No new data generated.

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
