# Peer review of "Orthobiologics Revisited: A Concise Perspective on Regenerative Orthopedics"

_cimb, 2025, doi:10.3390/cimb47040247_

Round 1

Reviewer 1 Report

Comments and Suggestions for Authors

This is a well-written review. The research topic is very interesting, and the writing style is fluent, concise, and logically structured. However, there are still some concerns that need to be addressed before publication:

  1. Introduction:
    There have been several reviews on orthobiologics in recent years. Since the title and emphasis are on a “revisited” perspective, could the authors clearly highlight the most critical innovations or refreshing viewpoints in this review in the introduction section compared to existing reviews?
  2. Methods:
    1. Have you included MEDLINE in your search strategy?
    2. Using MeSH terms in combination with free-text keywords might provide a more comprehensive search.
    3. How was the time frame (past decade) determined?
  3. Summary of the Mechanisms:
    1. I’m curious: how do stem cells remain active in these products? Do they still retain their self-renewal and multilineage differentiation capacities? Has any study tested this? The reference cited here (24) is also a narrative review. Are there any original studies that support these mechanisms?
    2. Since orthobiologics can contribute to different repair scenarios, how do the same stem cells/growth factors precisely differentiate? What are the regulatory mechanisms guiding tissue-specific repair? The mechanisms listed here are reasonable, but they are somewhat general and do not fully address this point of interest.
  4. Application:
    1. It would enhance clarity to include an additional table summarizing the applications of different orthobiologics across various clinical contexts.
    2. Page 7, line 280: This statement might not be entirely accurate—do ADSCs specifically tend to differentiate into adipocytes? BMSCs can also differentiate into similar lineages. What makes SVF-derived ADSCs particularly effective in this context?
    3. Page 7, line 287: Are there specific studies focused on muscle injury? The references cited here seem more general.
  5. Current Limitations:
    Have there been any large-scale, long-term clinical trials on orthobiologics to date? Are there any meta-analyses in this field? If so, the authors could consider including them as references in the manuscript (e.g., in addition to refs 38 and 51—are there others?).

Reviewer 2 Report

Comments and Suggestions for Authors

The manuscript entitled "Orthobiologics Revisited: A Concise Perspective on Regenerative Orthopedics" provides a comprehensive review of orthobiologics and their applications in regenerative medicine, particularly in orthopedics. The authors discuss different types of orthobiologics, their mechanisms of action, clinical applications, and future directions. However, there are several areas where the manuscript could be improved, particularly regarding the definition and scope of orthobiologics, as well as other aspects of the study.

The manuscript defines orthobiologics as "substances that occur naturally in the body and accelerate the healing of orthopedic injuries. While this definition is accurate, it is somewhat limited and does not fully capture the breadth of orthobiologics. It is important to note that orthobiologics can also be derived from allogeneic sources. The definition could be expanded to include exogenous biological substances such as recombinant growth factors and synthetic biomaterials used to enhance tissue repair and regeneration. Orthobiologics are not limited to autologous or naturally occurring substances; they may also include engineered or synthetic biologics designed to mimic or enhance natural healing processes. This broader definition would be consistent with the evolving landscape of regenerative medicine.

In addition, the tables in the manuscript provide limited information and could be improved or removed. Expanding the tables to provide more detailed and relevant data, or replacing them with more informative visual aids, would strengthen the overall quality of the manuscript and its usefulness to readers.

Reviewer 3 Report

Comments and Suggestions for Authors

I read with interest the review: Orthobiologics Revisited: A Concise Perspective on Regenerative Orthopedics.

The manuscript is well-written and provides a comprehensive overview of orthobiologics, addressing many key aspects of interest in regenerative orthopedics. However, I have a few suggestions for improvement. While the authors highlight the lack of well-conducted and reliable clinical studies as a major challenge in the field, a significant portion of the cited references originate from the authors themselves. To strengthen the manuscript's objectivity, I suggest including a dedicated section discussing the limitations of existing evidence and previous studies, or alternatively incorporating this discussion into an existing chapter.

Additionally, although ethical concerns have been addressed, the manuscript does not comprehensively discuss the potential side effects of orthobiologic treatments. For example, extracellular matrix (ECM) remodeling factors can be beneficial, but when their actions are uncontrolled, they may lead to tissue fibrosis, which could impair movement, restrict joint range of motion, or alter tissue mechanical properties, affecting contractility. Expanding on such risks for various treatment modalities would provide a more balanced perspective on their clinical application. Incorporating these aspects would further enhance the manuscript’s critical depth and completeness.

I believe implementing these minor revision the manuscript may improve.

Round 2

Reviewer 2 Report

Comments and Suggestions for Authors

All my concerns were addressed.